# A proteomic survival predictor for COVID-19 patients in intensive care

Vadim Demichev[1,2,3], Pinkus Tober-Lau[4], Tatiana Nazarenko[5,6], Oliver Lemke[1], Simran Kaur Aulakh[2], Harry J. Whitwell[7,8,9], Annika Röhl[1], Anja Freiwald[1], Mirja Mittermaier[4,10], Lukasz Szyrwiel[2], Daniela Ludwig[1], Clara Correia-Melo[2], Lena J. Lippert[4], Elisa T. Helbig[4], Paula Stubbemann[4], Nadine Olk[4], Charlotte Thibeault[4], Nana-Maria Grüning[1], Oleg Blyuss[11,12,13], Spyros Vernardis[2], Matthew White[2], Christoph B. Messner[1,2], Michael Joannidis[14], Thomas Sonnweber[15], Sebastian J. Klein[14], Alex Pizzini[15], Yvonne Wohlfarter[16], Sabina Sahanic[15], Richard Hilbe[15], Benedikt Schaefer[17], Sonja Wagner[17], Felix Machleidt[4], Carmen Garcia[4], Christoph Ruwwe-Glösenkamp[4], Tilman Lingscheid[4], Laure Bosquillon de Jarcy[4], Miriam S. Stegemann[4], Moritz Pfeiffer[4], Linda Jürgens[4], Sophy Denker[18], Daniel Zickler[19], Claudia Spies[20], Andreas Edel[20], Nils B. Müller[19], Philipp Enghard[19], Aleksej Zelezniak[2,21], Rosa Bellmann-Weiler[15], Günter Weiss[15], Archie Campbell[22,23], Caroline Hayward[24], David J. Porteous[22,23], Riccardo E. Marioni[22], Alexander Uhrig[4], Heinz Zoller[17], Judith Löffler-Ragg[15], Markus A. Keller[16], Ivan Tancevski[15], John F. Timms[6], Alexey Zaikin[5,6,8,25], Stefan Hippenstiel[4,26], Michael Ramharter[27], Holger Müller-Redetzky[4], Martin Witzenrath[4,26], Norbert Suttorp[4,26], Kathryn Lilley[3], Michael Mülleder[28], Leif Erik Sander[4,26], PA-COVID-19 Study group, Florian Kurth[4,27]‡*, Markus Ralser[1,2]‡

1 Charité–Universitätsmedizin Berlin, Department of Biochemistry, Berlin, Germany, 2 The Francis Crick Institute, Molecular Biology of Metabolism Laboratory, London, United Kingdom, 3 The University of Cambridge, Department of Biochemistry and Cambridge Centre for Proteomics, Cambridge, United Kingdom, 4 Charité–Universitätsmedizin Berlin, Department of Infectious Diseases and Respiratory Medicine, Berlin, Germany, 5 University College London, Department of Mathematics, London, United Kingdom, 6 University College London, Department of Women's Cancer, EGA Institute for Women's Health, London, United Kingdom, 7 National Phenome Centre and Imperial Clinical Phenotyping Centre, Department of Metabolism, Digestion and Reproduction, Imperial College London, London, United Kingdom, 8 Lobachevsky University, Laboratory of Systems Medicine of Healthy Ageing, Nizhny Novgorod, Russia, 9 Imperial College London, Section of Bioanalytical Chemistry, Division of Systems Medicine, Department of Metabolism, Digestion and Reproduction, London, United Kingdom, 10 Berlin Institute of Health, Berlin, Germany, 11 Lobachevsky University, Department of Applied Mathematics, Nizhny Novgorod, Russia, 12 University of Hertfordshire, School of Physics, Astronomy and Mathematics, Hatfield, United Kingdom, 13 Sechenov First Moscow State Medical University, Department of Paediatrics and Paediatric Infectious Diseases, Moscow, Russia, 14 Medical University Innsbruck, Division of Intensive Care and Emergency Medicine, Department of Internal Medicine, Innsbruck, Austria, 15 Medical University of Innsbruck, Department of Internal Medicine II, Innsbruck, Austria, 16 Medical University of Innsbruck, Institute of Human Genetics, Innsbruck, Austria, 17 Medical University of Innsbruck, Christian Doppler Laboratory for Iron and Phosphate Biology, Department of Internal Medicine I, Innsbruck, Austria, 18 Charité–Universitätsmedizin Berlin, Medical Department of Hematology, Oncology & Tumor Immunology, Virchow Campus & Molekulares Krebsforschungszentrum, Berlin, Germany, 19 Charité–Universitätsmedizin Berlin, Department of Nephrology and Internal Intensive Care Medicine, Berlin, Germany, 20 Charité–Universitätsmedizin Berlin, Department of Anesthesiology and Intensive Care, Berlin, Germany, 21 Chalmers University of Technology, Department of Biology and Biological Engineering, Gothenburg, Sweden, 22 University of Edinburgh, Centre for Genomic and Experimental Medicine, Institute of Genetics and Cancer, United Kingdom, 23 University of Edinburgh, Usher Institute, Edinburgh, United Kingdom, 24 University of Edinburgh, MRC Human Genetics Unit, Institute of Genetics and Cancer, Edinburgh, United Kingdom, 25 Centre for Analysis of Complex Systems, Sechenov First Moscow State Medical University, Moscow, Russia, 26 German Centre for Lung Research, Germany, 27 Bernhard Nocht Institute for Tropical Medicine, Department of Tropical Medicine, and University Medical Center Hamburg-Eppendorf, Department of Medicine, Hamburg, Germany, 28 Charité–Universitätsmedizin Berlin, Core Facility—High-Throughput Mass Spectrometry, Berlin, Germany



**Data Availability Statement:** The protein quantities table along with the associated metadata are provided in supplementary materials. All scripts used to train and assess the machine learning

models are likewise provided (S1 Data.zip). The mass spectrometry proteomics data have been deposited to the ProteomeXchange Consortium (http://proteomecentral.proteomexchange.org) via the PRIDE partner repository with the dataset identifier PXD029009. All scripts used to train and assess the machine learning models as well as the respective input proteomics data are provided in supplementary materials (S1 Data.zip).

**Funding:** This work was supported by the Berlin University Alliance (501_Massenspektrometrie, 501_Linklab, 112_PreEP_Corona_Ralser), by UKRI/NIHR through the UK Coronavirus Immunology Consortium (UK-CIC), the BMBF/DLR Projektträger (01KI20160A, 01ZX1604B, 01KI20337, 01KX2021), Charité-BIH Centrum für Therapieforschung (BIH_PA_covid-19_Ralser), the BBSRC (BB/N015215/1, BB/N015282/1), the Francis Crick Institute, which receives its core funding from Cancer Research UK (FC001134), the UK Medical Research Council (FC001134), and the Wellcome Trust (FC001134 and IA 200829/Z/16/Z), as well as the European Research Council (SyG 951475 to M.R.). The work was further supported by the Ministry of Education and Research (BMBF), as part of the National Research Node 'Mass spectrometry in Systems Medicine (MSCoresys), under grant agreements 031L0220A and 161L0221. The study was further supported by the German Federal Ministry of Education and Research (NaFoUniMedCovid19 – NUM -NAPKON, NUM-COVIM, FKZ: 01KX2021 and PROVID—FKZ: 01KI20160A) to Florian Kurth, Leif E. Sander, Martin Witzenrath, Norbert Suttorp and Stefan Hippenstiel. Leif Erik Sander is supported by the German Research Foundation (DFG, SFB-TR84 114933180) and by the Berlin Institute of Health (BIH), which receives funding from the Ministry of Education and Research (BMBF). Martin Witzenrath is supported by grants from the German Research Foundation, SFB-TR84 C06 and C09, by the German Ministry of Education and Research (BMBF) in the framework of the CAPSyS (01ZX1304B), CAPSyS-COVID (01ZX1604B), SYMPATH (01ZX1906A) and PROVID project (01KI20160A) and by the Berlin Institute of Health (CM-COVID). Stefan Hippenstiel is supported by the German Research Foundation (DFG, SFB-TR84 A04 and B06), and the BMBF (PROVID, and project 01KI2082). Norbert Suttorp is supported by grants from the German Research Foundation, SFB-TR84 C09 und Z02, by the German Ministry of Education and Research (BMBF) in the framework of the PROGRESS 01KI07114. The study was further supported by Wellcome Trust (200829/Z/16/Z). The Generation Scotland study received core support from the Chief Scientist Office of the

☙ These authors contributed equally to this work.
† Deceased.
‡ FK and MR also contributed equally to this work.
* florian.kurth@charite.de

# Abstract

Global healthcare systems are challenged by the COVID-19 pandemic. There is a need to optimize allocation of treatment and resources in intensive care, as clinically established risk assessments such as SOFA and APACHE II scores show only limited performance for predicting the survival of severely ill COVID-19 patients. Additional tools are also needed to monitor treatment, including experimental therapies in clinical trials. Comprehensively capturing human physiology, we speculated that proteomics in combination with new data-driven analysis strategies could produce a new generation of prognostic discriminators. We studied two independent cohorts of patients with severe COVID-19 who required intensive care and invasive mechanical ventilation. SOFA score, Charlson comorbidity index, and APACHE II score showed limited performance in predicting the COVID-19 outcome. Instead, the quantification of 321 plasma protein groups at 349 timepoints in 50 critically ill patients receiving invasive mechanical ventilation revealed 14 proteins that showed trajectories different between survivors and non-survivors. A predictor trained on proteomic measurements obtained at the first time point at maximum treatment level (i.e. WHO grade 7), which was weeks before the outcome, achieved accurate classification of survivors (AUROC 0.81). We tested the established predictor on an independent validation cohort (AUROC 1.0). The majority of proteins with high relevance in the prediction model belong to the coagulation system and complement cascade. Our study demonstrates that plasma proteomics can give rise to prognostic predictors substantially outperforming current prognostic markers in intensive care.

## Author summary

Healthcare systems around the world are struggling to accommodate high numbers of the most severely ill patients with COVID-19. Moreover, the pandemic creates a pressing need to accelerate clinical trials investigating potential new therapeutics. While various biomarkers can discriminate and predict the future course of disease for patients of different disease severity, prognosis remains difficult for patient groups with similar disease severity, e.g. patients requiring intensive care. Established risk assessments in intensive care medicine such as the SOFA or APACHE II show only limited reliability in predicting future disease outcomes for COVID-19. In this study we hypothesized that the plasma proteome, which reflects the complete set of proteins that are expressed by an organism and are present in the blood, and which is known to comprehensively capture the host response to COVID-19, can be leveraged to allow for prediction of survival in the most critically ill patients with COVID-19. Here, we found 14 proteins, which over time changed in opposite directions for patients who survive compared to patients who do not survive on intensive care. Using a machine learning model which combines the measurements of multiple proteins, we were able to accurately predict survival in critically ill patients with COVID-19 from single blood samples, weeks before the outcome, substantially outperforming established risk predictors.

Scottish Government Health Directorates (CZD/16/6) and the Scottish Funding Council (HR03006), and is now supported by the Wellcome Trust (216767/Z/19/Z). Archie Campbell is funded by HDR UK and the Wellcome Trust (216767/Z/19/Z). Caroline Hayward is supported by an MRC University Unit Programme Grant (MC_UU_00007/10) (QTL in Health and Disease). Riccardo Marioni is supported by an Alzheimer's Research UK project grant (ARUK-PG2017B-10). HJ. Whitwell, JF. Timms, A. Zaikin and T. Nazarenko are supported by a Medical Research Council grant (MR/R02524X/1) and HJ. Whitwell, A. Zaikin and O. Blyuss by the Ministry of Science and Higher Education agreement No. 075-15-2020-808. HJ. Whitwell is supported by the National Institute for Health Research (NIHR) Imperial Biomedical Research Centre (BRC). J. Timms is supported by the National Institute for Health Research (NIHR) UCLH/UCL Biomedical Research Centre. Mirja Mittermaier is a participant in the BIH-Charité Digital Clinician Scientist Program funded by the Charité – Universitätsmedizin Berlin, the Berlin Institute of Health, and the German Research Foundation (DFG). Markus A. Keller is supported by the Austrian Science Funds (FWF; P33333) and the Austrian Research Promotion Agency (FFG, #878654). The funders had no role in study design, data collection and analysis, decision to publish, or preparation of the manuscript.

**Competing interests:** The authors declare no competing interests. Author John F. Timms was unable to confirm their authorship contributions. On their behalf, the corresponding author has reported their contributions to the best of their knowledge.

# Introduction

The COVID-19 pandemic has brought health systems around the globe to the brink of collapse. Capacities for intensive care treatment of patients with organ failure have reached their limits in many regions with intense SARS-CoV-2 transmission and were often central to political decisions regarding restrictions on public life, e.g. through contact restrictions or lockdowns. The global impact of the pandemic increases the pressures to devise new clinical approval strategies so that potential therapeutics can be identified and tested faster, at higher accuracy, and in clinical trials with smaller sample sizes [1]. Various models for classification of disease severity and for prediction of clinical trajectories and outcome have been developed for COVID-19, based on laboratory measurements, clinical scores, imaging, and omics technologies [2–5]. These pointed to the importance of specific immune cells, inflammatory and antiviral cytokines and chemokines, as well as the coagulation cascade in COVID-19 disease progression [5–13]. They predict the risk of the future need for mechanical ventilation in the heterogeneous group of patients at early time points, e.g. at admission to the hospital, when clinical parameters and biomarkers differ substantially between mildly affected and severely ill patients [2–5,14].

Treatment decisions within the most severely ill patients, for instance whether a patient should be treated with extracorporeal membrane oxygenation (ECMO), have a major impact on resources. Currently, such decisions are often based primarily on the patient's age, comorbidities, and established intensive care prognosis models, such as the Sequential Organ Failure Assessment (SOFA) or Acute Physiology and Chronic Health Evaluation (APACHE II), which assess the patient on the basis of a combination of established clinical and laboratory risk parameters [15,16]. However, the predictive values of both SOFA and APACHE II for the most critical forms of COVID-19 are limited [17–19], creating a diagnostic gap and imminent need for reliable predictors, specifically validated in severely ill COVID-19 patients, to guide and tailor efforts in treating these critically ill patients. Moreover, the lack of reliable predictors increases the challenge of interpreting the results of early phase clinical trials, which typically enroll low numbers of patients. Indeed, testing for the success of a clinical intervention requires classification of divergent clinical trajectories within more homogeneous groups, such as WHO grade 7 patients. At least in COVID-19, this is hampered by the fact that molecular signatures within a group of patients with comparable disease severity are considerably more similar when compared to the differences between mild and severe patients [6,7,14].

Plasma proteomics holds the promise of integrating the genetic background of an individual with their life history, physiological, nutritional, and demographic parameters, and hence, have the potential to form the foundation of a new generation of predictors [20–24]. Among the spectrum of proteomic technologies available, mass spectrometry has the appeal that once markers are identified, they allow for the direct generation of targeted panel assays measurable by selective reaction monitoring (SRM), simplifying their implementation into clinical routine. Recently, new mass spectrometry based proteomic technologies have been developed to increase throughput and measurement precision, so that the path from discovery to application is simplified [6,25–28].

We studied proteomes of two well characterized cohorts of the most severely ill patients with COVID-19 in two independent health care centers (Charité–Universitätsmedizin Berlin, Germany, and Medical University of Innsbruck, Austria) who gave informed consent to deep clinical and molecular phenotyping [14,19,29]. Using a recently published dataset from our group [14], we specifically assessed whether proteomic measurements can be used to predict the outcome (death vs. survival) of severe COVID-19 from time series data,

as well as from samples taken at key clinical decision points. We found 14 protein concentration trajectories that, over the timeline of disease progression, distinguish survivors from non-survivors. Moreover, a machine learning (ML) model, based on parenclitic networks, generated accurate prognosis on single time point samples that were collected once the patient reached the maximum treatment level. Emphasizing the prognostic potential of the proteome, this sample was, in median, taken 39 days before outcome. The ML predictor trained on these samples substantially outperformed established clinical risk scores and predicted the outcome among a group of severely ill patients with similar clinical presentation with high accuracy.

## Results

The exploratory cohort used for marker identification and model generation consisted of the 50 most severely ill COVID-19 patients out of a cohort of 168 patients with varying disease severity, treated between 15 March and 16 September 2020 at Charité University Hospital, Berlin, Germany, a tertiary care referral centre for the treatment of ARDS with associated weaning centre (**Fig 1A**) [14,19,29]. There were no treatment restrictions due to shortages of intensive care capacity at the time of this patient cohort. The 50 patients selected for the study were treated in intensive care with invasive mechanical ventilation plus additional organ support such as renal replacement therapy (RRT), ECMO, or vasopressors, corresponding to grade 7 on the WHO Ordinal Scale for Clinical Improvement. Patients with limitations of therapy according to their wish were excluded. Thirty-six (72%) patients required RRT, 19 (38%) patients were treated with ECMO, and 16 (32%) patients were treated with both RRT and ECMO. Fifteen (30%) patients died. Median time of hospitalization in survivors was 63 days (n = 35, IQR 44–89). Median time from admission to death was 28 days (n = 15, IQR 16–43). Patient characteristics are shown in S1 Table. The details on the proteomic workflow, protein detection rates, as well as patient trajectories are provided in S1 and S2 Figs of our previous work [14].

Within this treatment group of critically ill COVID19 patients, the Charlson Comorbidity Index [30,31] performed poorly in classifying survivors from non-survivors by AUROC values of 0.63 (P = 0.16, **Fig 2A**). From a time-resolved data resource for the PA-COVID-19 study, spanning over a compendium of clinical parameters, plasma proteomes, cell counts, enzyme activities, and outcomes [14], we further determined the SOFA and APACHE II scores. These scores, too, could not confidently distinguish survivors from non-survivors (**Fig 2A**, AUROC = 0.68, P = 0.05 for APACHE II score at ICU admission, and AUROC = 0.65, P = 0.11 for SOFA score at the time of first sampling at WHO grade 7).

Studying the plasma proteomes [14] we found 78 proteins for which the concentration changed significantly during the patients' disease course. Out of these proteins, 14 were found to change differently over time for survivors and non-survivors (**Fig 1B, Fig 1C**). Patients with fatal outcomes were characterized by a significant increase in inflammatory proteins over time (SAA1, SAA2, CRP, ITIH3, LRG1, SERPINA1, SERPINA10 and LBP). Conversely, the levels of these proteins in plasma decreased over time in survivors. Moreover, anti-inflammatory proteins (SERPINA4, A2M) decreased over time in non-survivors, indicating a persistent pro-inflammatory signature. Similarly, two key proteins of the coagulation system, thrombin (F2) and plasma kallikrein (KLKB1), known to be decreased in severe COVID-19 [12,14], further decreased over time in non-survivors, while increasing in survivors.

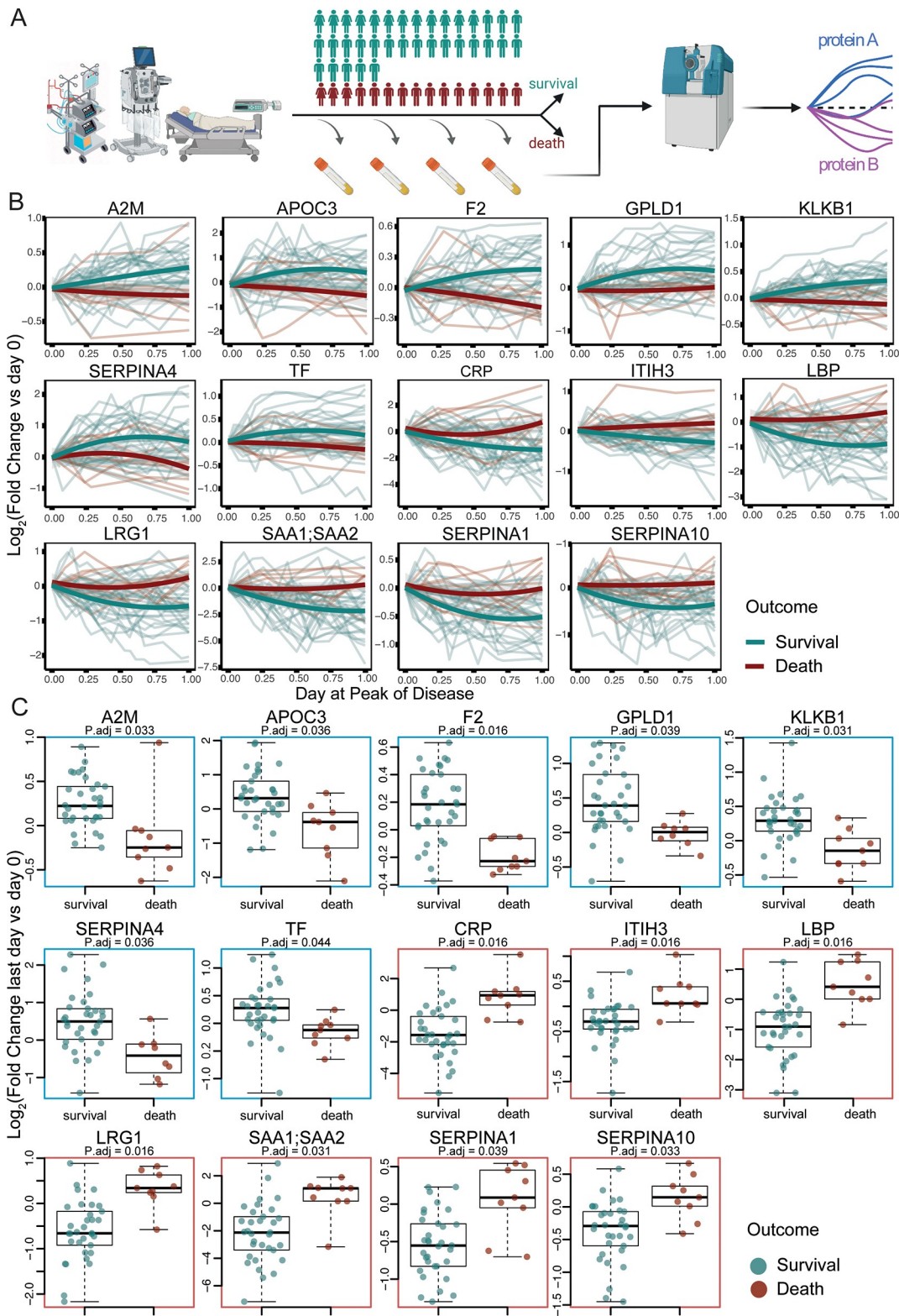

**Fig 1. Protein concentration trajectories that differentiate survivors of critical COVID-19 from non-survivors. a)** Fifty Patients with PCR-confirmed COVID-19 treated at Charité University Hospital Berlin, Germany, were sampled longitudinally, to generate high-resolution time series for 321 protein quantities. In parallel, precise clinical phenotyping was performed, including recording of intensive care and disease severity scores, treatment parameters, and outcome

(PA-COVID-19 data resource [14]). **b)** Protein level trajectories over time (FDR < 0.05), for which time-dependent concentration changes (y-axis: log2 fold change) during the peak of the disease differentiate survivors from non-survivors in critically ill patients (Methods). **c)** as **b)** but expressed as boxplots (log2 fold change last vs first day). Figure created with BioRender.com.

For diagnostic purposes and treatment decisions time series data is however impractical to obtain. We therefore explored the potential of using single time point samples to predict outcome. We chose the earliest sample obtained after the critical decision regarding escalation of treatment, i.e. the earliest sample obtained at the maximum treatment level (WHO grade 7), to generate an outcome predictor. The median time from sampling until the outcome was 39 (IQR 16–64) days in our cohort. Using 57 proteins for which targeted mass spectrometric assays (MRM assays) as listed in the MRMAssayDB [32] are available, indicating that they have been selected for a clinical or biomedical indication also in other context, we established a machine learning model based on parenclitic networks, a graph-based approach in which networks representing the deviation of an individual from the population are derived [33,34]. The networks are generated by considering every pair of analytes (proteins) individually and calculating the respective edge weight as the estimated probability of fatal outcome based on this pair of proteins. Predictive models are then generated by considering the topological differences between networks from individual cases (non-survivors vs. survivors) (Methods). We achieved high prediction accuracy on the test subjects, who were excluded when training the machine learning model (in a cross-validation fashion, see Methods), with AUC = 0.81 (95% CI 0.68–0.94) for the receiver-operating characteristic (ROC) curve (**Fig 2B**). Out of the 25 proteins with the highest relevance in the parenclitic model, 15 are components of the coagulation system and 8 proteins belong to the complement cascade (**S2 Table**). To further demonstrate that the proteomic data contains sufficient physiological information to allow outcome prediction, i.e. that the results are not restricted to a specific algorithm, we also tested a model based on a support vector machine (SVM). The SVM proved to be capable of survival prediction as well, albeit with inferior performance compared to the parenclitic network (**S1 Fig**).

To independently validate the potential of the plasma proteome to predict outcomes in critically ill COVID19 patients, we examined the performance of the parenclitic network trained on our prime cohort (Charité) on an independent cohort of 24 patients with critical COVID-19 from Austria (survival n = 19, death n = 5, median time between sampling and outcome 22 days, interquartile range 15–42 days) ('Innsbruck' cohort, Methods). Despite the validation cohort originating from a different hospital and health care system, the machine learning model demonstrated high predictive power on this independent cohort (AUROC = 1.0, P = 0.000047, **Fig 2C**). Using the cutoff value for survival prediction derived from the Charité cohort, the model correctly predicted the outcome for 18 out of 19 patients who survived and for 5 out of 5 patients who died in this independent 'Innsbruck' cohort.

## Discussion

The prognostic value of several biomarkers (e.g. CRP, IL-6, ferritin) and clinical scores for predicting disease progression in COVID-19 at early disease stages, e.g. at hospital admission, is now well established [35,36]. For the comparatively homogeneous subgroup of severely ill patients already requiring mechanical ventilation and additional organ support, prediction of future disease trajectories and outcome (survival or death) is by far more challenging, and only limited data exist [17,37,38]. Moreover, clinical severity scores are often not validated for unconscious patients, and laboratory measurements are frequently confounded by intensive care treatment. Outcome of ICU patients may further be critically determined by resource

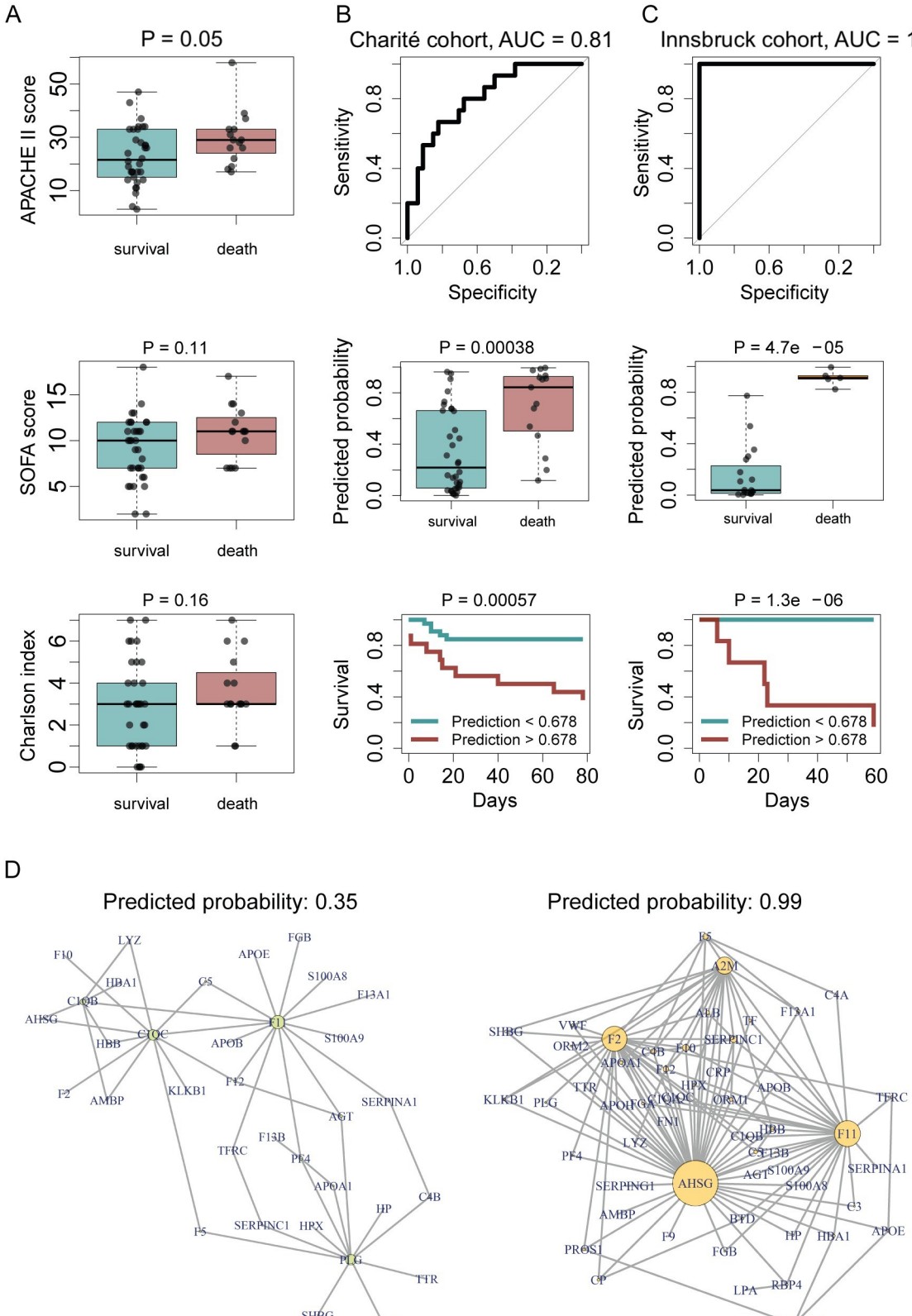

**Fig 2. Prediction of survival or death in critically ill patients, from the first sampling time point at intensive care treatment level (WHO grade 7). a)** Performance of established ICU risk assessment indices (APACHE II, SOFA and Charlson comorbidity

index) calculated at the time of ICU admission (APACHE II, Charlson comorbidity index) or at the first time point at WHO grade 7 (SOFA score) in predicting the outcome in critically ill patients. **b)** Prediction of survival or death in critically ill patients using proteomics. A machine learning model based on parenclitic networks (Methods) was trained on the samples of the Charité cohort closest to the time point of treatment escalation during intensive care (start of ECMO, RRT or vasopressors, i.e. WHO grade 7). The performance was assessed on the test samples, which were held out during training. **Upper panel:** The ROC curve indicates correct classification of survival vs non-survival with an AUROC of 0.81 (95% CI 0.68–0.94). **Middle panel**: The proteomic classifier was used to predict the probability of survival and non-survival, which is significantly different between the groups. **Lower panel**: Kaplan-Meier survival curves using a threshold of predicted probability (0.678) chosen to maximize Youden's J index (J = sensitivity + specificity—1). Log-rank test was used to compare survival rates between patients with predicted death risk < 0.678 (black) and > 0.678 (orange). **c) (upper, middle, and lower panels):** The model trained on the Charité cohort, was tested on an independent cohort (Innsbruck). **d)** Exemplary parenclitic networks from two patients in the independent Innsbruck cohort. Edges with weights > 0.5 are shown. Left panel: a network predicting low probability of death in a surviving patient. Right panel: a network predicting high probability of death in a non-survivor.

constraints, the varying level of experience with organ replacement therapies or the rates of superinfection, rendering prediction complex [38]. On the other hand, patients in intensive care units, and particularly those in need of special organ replacement therapies such as ECMO, require a disproportionately large share of resources compared to other patients, so decisions to initiate such therapies should be based on the best information and assessment possible. Prognostic tools in critically ill patients are hence of crucial importance to guide and tailor the treatment efforts. This is particularly true in a situation when health care systems are overstrained. Another key potential for the use of outcome predictors is clinical trial monitoring, where measurements of prognostic molecular signatures over time can be used to evaluate experimental therapies on an individual, time-resolved basis. Moreover, an accurate outcome predictor would allow us to test whether a given treatment changes the predicted trajectory of an individual patient.

Previously, we and others investigated plasma proteome alterations in COVID-19 [6–8,10,12,14], which show a remarkable ability to classify the severity of disease. For instance, our investigations showed that the host response in the early inflammatory phase creates a strong signature in the plasma proteome, and is critical as well as predictive about the future disease progression in severe COVID-19 [14]. New proteomic platform technologies have significantly gained precision and throughput compared to their predecessors, rendering the application of multivariate regression models more effective and bringing them increasingly close to routine clinical use [6]. Importantly, even without platform technologies, biomarkers identified in proteomic profiles can be translated into clinical use, e.g. by using standard techniques such as selective reaction monitoring (SRM) for the quantification of protein panels, or enzyme linked immunosorbent assays (ELISA) for the sensitive quantification of individual biomarkers.

Here, we show that an increase in specific inflammatory and acute phase proteins over time (e.g., SAA1;SAA2, CRP, ITIH3, LRG1, SERPINA1, and LBP) is associated with the risk of death from COVID-19, while an increase of kallikrein (KLKB1), kallistatin (SERPINA4), thrombin (F2), apolipoprotein C3 (APOC3), GPLD1, and the protease inhibitor A2M, is associated with survival. Interestingly, we and others have found all of these proteins to also be differentially expressed depending on disease severity in COVID-19 [6,7,10,12,14]. Moreover, there is substantial overlap with a panel of proteins predictive of mortality in COVID-19 identified by Völlmy et al. [39]. Hence, despite only a subset of proteins that are differentially concentrated depending on disease severity predict outcome, and the fact that typical single-centre ICU studies are conducted on small numbers of patients, this result indicates a high congruence and reproducibility of plasma proteome signatures across studies.

SAA1;SAA2, CRP, ITIH3, SERPINA1 are acute phase proteins that are also dysregulated in other inflammatory states including sepsis [40]. Increased LRG1 and LBP as well as decreased A2M [40,41] are indicators of an ongoing immune response, complementing

the general pro-inflammatory signature of these predictive proteins. APOC3 and GPLD1 are involved in lipid metabolism, which has been shown to be dysregulated in bacterial pneumonia, thereby associated with unfavorable outcomes [42]. Kallikrein is involved in the blood coagulation system, fibrinolysis, and the complement cascade, three systems known to be dysregulated in COVID-19 [43–45]. It mediates the cleavage of kininogen to bradykinin and des-Arg$^9$-bradykinin, a potent vasoactive peptide which is counter-regulated by ACE2, the cell entry receptor for SARS-CoV-2. Since the loss of ACE2 in COVID-19 supposedly leads to an imbalance of bradykinins, inhibition of the kallikrein-kinin system has been discussed as a treatment strategy in COVID-19 [46–48]. This hypothesis is not supported by our data, which indicate improved prognosis with increasing kallikrein levels. Kallikrein is counterbalanced by kallistatin, which equally increased over time in survivors in our study population, thereby potentially equilibrating the increase in the kinin-kallikrein system. Kallistatin is known for pleiotropic effects in vascular repair, endothelial function, and inflammation [49] and possesses protective properties in acute lung injury. According to our data kallistatin should be considered as a potential candidate for clinical testing in critical COVID-19 [50].

While prognostic assessments based on repeated measurements over time allow for treatment monitoring, including evaluation of experimental therapies in clinical trials, prognostic measurements from single time points are particularly valuable for timely patient management and resource allocation. We therefore employed a machine learning model to integrate proteomic measurements from the first time point at WHO grade 7, i.e. invasive mechanical ventilation and additional organ support therapy, in order to derive prognosis of outcome. We achieve high prognostic values, both in the exploratory cohort, as well as in a fully independent cohort.

The results are currently based on a comparatively small number of patients with adverse outcome. Given the naturally small sample sizes of ICU cohorts and the exploratory character of our study, findings will have to be validated in larger cohorts, before further steps can be undertaken to translate our findings into clinical practice in the future. The panel of proteins identified in our study should also be assessed for other conditions such as non-COVID-19 ARDS.

The majority of proteins with the highest relevance for the machine learning predictor were components of the coagulation system and the complement cascade (**S2 Table**). Both systems are known to be crucial for treatment and disease courses for severely ill COVID-19 patients [9,10]. This is particularly well illustrated by recent data from a multi-platform clinical trial indicating that a substantial proportion of patients with severe COVID-19 develop thrombo-embolic events despite therapeutic anticoagulation [51,52]. The protein with the highest relevance in our model is Fetuin-A (AHSG), which is known to be strongly downregulated in severe COVID-19 [10,14]. Of note, genetic polymorphisms associated with higher AHSG plasma concentrations were found to be protective in SARS-CoV-1 infection [53]. One important function of AHSG is regulation of inflammation through deactivation of macrophages [54], and there is emerging evidence that macrophages play a key role in pulmonary inflammation and dysfunction in COVID-19 [11,55–57]. A number of proteins identified as outcome predictors have also been shown to be differentially expressed in sepsis, including SAA1, CRP, SERPINA1, KLKB1, and A2M [40], indicating a general inflammatory signature rather than specific markers of COVID-19.

In summary, we have leveraged the power of the proteome to address a problematic diagnostic gap in the prognosis of the most critical form of COVID-19, that is not covered by established clinical assessments, such as the SOFA or APACHE II scores. We show that the proteome accurately predicts survival in critically ill patients with COVID-19, from samples

that were collected 39 days in median before the outcome. The findings warrant further prospective assessment of proteomic predictors and the described models in larger cohorts. The majority of proteins with high relevance in the model are components of the coagulation system and complement cascade, highlighting their critical role in progression and outcome of most severe COVID-19.

## Methods

### Charité patient cohort and clinical data

Patients included in this analysis are a sub-cohort of the PA-COVID-19 study conducted at Charité—Universitätsmedizin Berlin, a prospective observational cohort study on the pathophysiology of COVID-19 as described previously [14,19,29]. All patients with PCR-confirmed SARS-CoV-2 infection that progressed to critical disease (WHO grade 7, i.e. invasive mechanical ventilation and additional organ support), were eligible for inclusion. Exclusion criteria included refusal to provide informed consent by the patient or a legal representative, and any condition prohibiting serial biosampling. Patients were treated according to current clinical guidelines. Patients for whom limitation of therapy was decided according to the patient's wish were excluded from analysis. This includes three cases, for whom limitation of therapy was decided at a later time point according to the patient's presumed wish and predictably unfavorable outcome. All other patients received maximum intensive care treatment including organ replacement therapies at the discretion of the responsible physicians. One patient (ID 135), who was still hospitalized and clinically improving 5 months after admission, was classified as a survivor. One patient still in critical condition 5 months after admission was excluded due to uncertain outcome.

Biosampling of EDTA plasma for proteome measurement was performed up to 3 times per week after inclusion. Disease severity was assessed according to the WHO ordinal scale for clinical improvement (World Health Organisation 2020). Clinical data were captured in secuTrial (interActive Systems GmbH, Berlin, Germany). Pseudonymized data exported from secuTrial were processed using JMP Pro 15 (SAS Institute Inc., Cary, NC, USA).

### Innsbruck Patient cohort and clinical data

Serum samples from patients admitted to the intensive care unit at the Department of Medicine, University Hospital of Innsbruck with PCR-confirmed severe COVID-19 were collected within the first days (median 7.5, IQR 5–12) after admission, and written informed consent was obtained. Patients were treated according to national guidelines. The study was approved by the local ethics research committee EK-Nr. 1107/2020, and EK-Nr. 1103/2020 for follow-up.

### Statistical analysis and multiple-testing correction

Statistical testing on proteomic and diagnostic data was performed in the R environment for statistical computing, version 3.6.0 [58], as described previously [14]. Briefly, all protein measurements were first log2-transformed and only protein groups matched to at least three different peptides were considered. Quantities of gene products corresponding to open reading frames IGxx (i.e. different types of immunoglobulin chains) were summed together to generate quantities representative of the overall levels of immunoglobulin classes (IGHVs, IGLVs, etc). Imputation of missing data was not performed. Significance testing for equal medians was performed using the Mann-Whitney U test, as implemented in the "wilcox.test" function of the "stats" R package. A non-parametric test was chosen here to minimise the influence of outliers on the calculated p-values. Multiple-testing correction

was performed using the Benjamini-Hochberg false discovery rate controlling procedure [59], implemented in the "p.adjust" function of the "stats" R package. Adjusted p-values below 0.05 were considered significant.

### Identifying omics trajectories that are predictive of survival at the peak period of the disease

For each omics feature, the difference between its log2-levels at the last and the first sampling timepoints during the peak period of the disease was considered. This period was defined as the time when the patient was receiving the most intensive treatment during their stay in hospital, that is the time when the patient was at WHO grade 6 or 7. The distribution of this difference between survivors and non-survivors was compared using the Mann-Whitney U test. Only non-DNI patients with known outcome were included.

### Prediction of survival

The first time point measured at the WHO grade 7 was selected per patient, to train the survival predictor. This ensured that 'future' information, encoded in the later time points, was not used for predictor training. To reduce the feature space used as input for the machine learning model, we limited it to the quantities of 57 proteins which are FDA-approved biomarkers with MRM assays available [32] and which were quantified with at least three different peptides in this study. Missing values were imputed using minimal value imputation, and the data were standardized.

Machine learning was carried out using the parenclitic networks approach [33,34]. Briefly, during training, for each pair of features, a radial SVM classifier is trained (using the svm() function from the "e1071" R package with default settings). For each sample, a network is then built, wherein vertices correspond to features and the edge weight is the death probability as predicted by the SVM classifier. Maximum, mean and standard deviation of the edge weights, as well as the numbers of edges with weights greater than 0.5 (i.e. fatal outcome is predicted) and nodes with at least one such edge are calculated. A LASSO classification model (alpha = 0.01) is then constructed on these 5 features using the glmnet() function of the "glmnet" [60] R package with default settings.

For the assessment of the classifier performance (Charité cohort), a cross-validation method was applied in the following way: the prediction was made for each sample by excluding (withholding) it from the dataset along with two other samples (chosen randomly with the constraint that out of 3 samples one corresponds to a non-survivor and two to survivors), training the classifier on the remaining (independent) samples and then generating predictions for the withheld samples using the trained model. Such a leave-3-out partition was generated randomly 50 times and the predictions for each sample were averaged. The partitioning strategy ensured that the evaluation of the predictive performance would not be affected by any potential overfitting, no matter how significant. For the assessment of the performance on an independent dataset (Innsbruck cohort), the classifier was trained on all the Charité samples and used to estimate the probabilities of fatal outcome on the Innsbruck cohort. The source code is provided in supplementary materials.

The 'relevance' scores for proteins in the parenclitic model were calculated as Kleinberg's authority centrality scores for the respective vertices in the "generalizing network". This network was generated by (i) replacing edge weights greater than 0.5 with 1.0 and weights less than 0.5 with 0.0 in the networks corresponding to non-survivors and (ii) averaging the resulting networks.

For survival prediction using support vector machines (SVM), the same data and the same selection of proteins as for the parenclitic network model was applied. The SVM was built in Python 3.8.5 using the SVC() function with an rbf-kernel and a gamma value of 0.005 as implemented in scikit-learn 0.23.2 [61]. To circumvent class-imbalances, balanced class-weights were assumed. For benchmarking the model a stratified 10-fold cross-validation was performed. The data were scaled to zero mean and unit variance based on the training data. The reported results for the Charité-cohort are based on the data that were withheld when constructing the model in each cross-validation step. For validating the model, a model was trained on all samples of the Charité-cohort and validated using the independent Innsbruck-cohort. p-Values were calculated using the Mann-Whitney U test as implemented in SciPy 1.5.2 [62]. AUC values and confidence intervals were obtained using the roc() function of the pROC R package.

We followed the guidelines for transparent reporting of multivariable prediction models for individual prognosis or diagnosis (TRIPOD) as proposed by the EQUATOR network [63].

## Study approval

The study was approved by the ethics committee of Charité—Universitätsmedizin Berlin (EA2/066/20) and conducted in accordance with the Declaration of Helsinki and guidelines of Good Clinical Practice (ICH 1996). Written informed consent was obtained from all patients or legal representatives according to regulations set by the ethics committee of Charité—Universitätsmedizin Berlin. The study is registered in the German and the WHO international registry for clinical studies (DRKS00021688).

## Supporting information

**S1 Tripod Checklist. Transparent reporting of multivariable prediction models for individual prognosis or diagnosis (TRIPOD) checklist as proposed by the EQUATOR network [63].** Checklist includes location of key aspects within the manuscript.
(PDF)

**S1 Table. Baseline, treatment, and outcome characteristics of patient cohort with severe COVID-19 receiving maximum therapy at Charité—University hospital Berlin.**
(DOCX)

**S2 Table. Top 25 proteins included in the machine learning model, ordered by their estimated 'relevance' scores (Methods).** Red writing indicates proteins involved in the complement system. Blue writing indicates proteins involved in coagulation.
(DOCX)

**S1 Fig. Performance of an SVM model in predicting survival for critical (WHO grade 7) COVID-19 patients.** Left panel: Boxplot of the decision function of the SVM for the Charité cohort. Displayed is the performance on the test data that were not used for model training. Middle panel: Boxplot of the decision function of the SVM for the Innsbruck cohort using a pre-trained model based on the Charité-cohort. Right panel: ROC-Curve and AUC corresponding to the boxplots for the Charité-cohort (black) and for the Innsbruck-cohort (red). AUC values of 0.66 (95% CI 0.49–0.84) and 0.88 (95% CI 0.67–1.0) were obtained for the Charité and Innsbruck cohorts, respectively.
(EPS)

**S1 Data. Scripts used to train and assess machine learning models as well as the respective input proteomics data.**
(ZIP)

## Acknowledgments

We thank Jan-David Manntz (Beckman, Germany) for help with the Biomek i7, Robert Lane, Jean-Baptiste Vincedent and Nick Morrice (SCIEX) for help with the TripleTOF 6600.

## PA-COVID-19 Study group, Charité–Universitätsmedizin Berlin

Malte Kleinschmidt, Katrin M. Heim, Belén Millet, Lil Meyer-Arndt, Nils B. Müller, Ralf H. Hübner, Tim Andermann, Jan M. Doehn, Bastian Opitz, Birgit Sawitzki, Daniel Grund, Peter Radünzel, Mariana Schürmann, Thomas Zoller, Fridolin Steinbeis, Florian Alius, Philipp Knape, Astrid Breitbart, Yaosi Li, Felix Bremer, Panagiotis Pergantis, Susanne Fieberg, Anne Wetzel, Moritz Müller-Plathe, Timur Özkan, Carola Misgeld, Dirk Schürmann, Bettina Temmesfeld-Wollbrück, Britta Stier, Martin Möckel, Jan A. Graaw, Victor Wegener, Marc Kastrup, Felix Balzer, Daniel Wendisch, Sophia Brumhard, Sascha S. Haenel, Philipp Georg, Claudia Conrad, Kai-Uwe Eckardt, Lukas Lehner, Jan M. Kruse, Carolin Ferse, Roland Körner, Andreas Edel, Steffen Weber-Carstens, Alexander Krannich, Saskia Zvorc, Linna Li, Uwe Behrens, Sein Schmidt, Maria Rönnefarth, Christina Pley, Claudia Fink, Chantip Dang-Heine, Robert Röhle, Emma Lieker, Christian Wollboldt, Yinan Wu, Georg Schwanitz, Constanze Lüttke, Denise Treue, Michael Hummel, Victor M. Corman, Christian Drosten, Christof von Kalle

## Author Contributions

**Conceptualization:** Leif Erik Sander, Florian Kurth, Markus Ralser.

**Data curation:** Vadim Demichev, Pinkus Tober-Lau, Lukasz Szyrwiel, Lena J. Lippert, Elisa T. Helbig, Paula Stubbemann, Nadine Olk, Charlotte Thibeault, Matthew White, Christoph B. Messner, Michael Joannidis, Thomas Sonnweber, Sebastian J. Klein, Alex Pizzini, Yvonne Wohlfarter, Sabina Sahanic, Richard Hilbe, Benedikt Schaefer, Sonja Wagner, Felix Machleidt, Carmen Garcia, Christoph Ruwwe-Glösenkamp, Tilman Lingscheid, Laure Bosquillon de Jarcy, Miriam S. Stegemann, Moritz Pfeiffer, Linda Jürgens, Sophy Denker, Daniel Zickler, Claudia Spies, Philipp Enghard, Rosa Bellmann-Weiler, Günter Weiss, Alexander Uhrig, Heinz Zoller, Judith Löffler-Ragg, Markus A. Keller, Ivan Tancevski, Holger Müller-Redetzky, Martin Witzenrath, Norbert Suttorp, Michael Mülleder.

**Formal analysis:** Vadim Demichev, Pinkus Tober-Lau, Tatiana Nazarenko, Oliver Lemke, Simran Kaur Aulakh, Harry J. Whitwell, Annika Röhl, Mirja Mittermaier, Lukasz Szyrwiel, Charlotte Thibeault, Nana-Maria Grüning, Oleg Blyuss, Spyros Vernardis, Matthew White, Christoph B. Messner, Aleksej Zelezniak, Archie Campbell, Caroline Hayward, David J. Porteous, Riccardo E. Marioni, John F. Timms, Alexey Zaikin, Stefan Hippenstiel, Michael Ramharter, Kathryn Lilley, Michael Mülleder.

**Investigation:** Vadim Demichev, Pinkus Tober-Lau, Oliver Lemke, Harry J. Whitwell, Annika Röhl, Anja Freiwald, Mirja Mittermaier, Daniela Ludwig, Clara Correia-Melo, Lena J. Lippert, Elisa T. Helbig, Paula Stubbemann, Charlotte Thibeault, Christoph B. Messner, Tilman Lingscheid, Claudia Spies, Andreas Edel, Nils B. Müller, Michael Ramharter, Kathryn Lilley, Michael Mülleder, Leif Erik Sander, Florian Kurth, Markus Ralser.

**Methodology:** Leif Erik Sander, Florian Kurth, Markus Ralser.

**Resources:** Harry J. Whitwell, Oleg Blyuss, Archie Campbell, Caroline Hayward, Riccardo E. Marioni, Markus A. Keller, John F. Timms, Alexey Zaikin, Stefan Hippenstiel, Martin Witzenrath, Norbert Suttorp.

**Supervision:** Leif Erik Sander, Florian Kurth, Markus Ralser.

**Validation:** Vadim Demichev.

**Visualization:** Vadim Demichev, Pinkus Tober-Lau, Simran Kaur Aulakh, Michael Mülleder, Leif Erik Sander.

**Writing – original draft:** Vadim Demichev, Pinkus Tober-Lau, Leif Erik Sander, Florian Kurth, Markus Ralser.

**Writing – review & editing:** Vadim Demichev, Pinkus Tober-Lau, Oliver Lemke, Simran Kaur Aulakh, Harry J. Whitwell, Annika Röhl, Anja Freiwald, Mirja Mittermaier, Lukasz Szyrwiel, Daniela Ludwig, Clara Correia-Melo, Lena J. Lippert, Elisa T. Helbig, Paula Stubbemann, Nadine Olk, Charlotte Thibeault, Oleg Blyuss, Spyros Vernardis, Matthew White, Christoph B. Messner, Michael Joannidis, Thomas Sonnweber, Sebastian J. Klein, Alex Pizzini, Yvonne Wohlfarter, Sabina Sahanic, Richard Hilbe, Sonja Wagner, Felix Machleidt, Carmen Garcia, Christoph Ruwwe-Glösenkamp, Tilman Lingscheid, Laure Bosquillon de Jarcy, Miriam S. Stegemann, Moritz Pfeiffer, Linda Jürgens, Sophy Denker, Daniel Zickler, Claudia Spies, Andreas Edel, Nils B. Müller, Philipp Enghard, Aleksej Zelezniak, Rosa Bellmann-Weiler, Günter Weiss, Archie Campbell, Caroline Hayward, David J. Porteous, Riccardo E. Marioni, Alexander Uhrig, Heinz Zoller, Ivan Tancevski, John F. Timms, Alexey Zaikin, Stefan Hippenstiel, Michael Ramharter, Holger Müller-Redetzky, Martin Witzenrath, Norbert Suttorp, Kathryn Lilley, Michael Mülleder, Leif Erik Sander, Florian Kurth, Markus Ralser.

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
