## [Decision Letter · Decision Letter 0]

27 Aug 2021

PDIG-D-21-00015A proteomic survival predictor for COVID-19 patients in intensive carePLOS Digital Health

Dear Dr. Kurth,

Thank you for submitting your manuscript to PLOS Digital Health. After careful consideration, we feel that it has merit but does not fully meet PLOS Digital Health’s publication criteria as it currently stands. Therefore, we invite you to submit a revised version of the manuscript that addresses the points raised during the review.

We look forward to receiving your revised manuscript.

Kind regards,

Reinhard Bauer

Guest Editor

PLOS Digital Health

Journal Requirements:

1. Please amend your detailed Financial Disclosure statement. This is published with the article, therefore should be completed in full sentences and contain the exact wording you wish to be published.

i). State what role the funders took in the study. If the funders had no role in your study, please state: “The funders had no role in study design, data collection and analysis, decision to publish, or preparation of the manuscript.”

2. Please provide separate figure files in .tif or .eps format only, and remove any figures embedded in your manuscript file.  If you are using LaTeX, you do not need to remove embedded figures.

For more information about figure files please see our guidelines: https://journals.plos.org/digitalhealth/s/figures

3. We notice that your supplementary tables are included in the manuscript file. Please remove them and upload them  with the file type 'Supporting Information'. Please ensure that all Supporting Information files are included correctly and that each one has a legend listed in the manuscript after the references list.

4. We do not publish any copyright or trademark symbols that usually accompany proprietary names, eg (R), (C), or TM  (e.g. next to drug or reagent names). Therefore please remove all instances of trademark/copyright symbols throughout the text, including SecuTrial® on page 14.

Additional Editor Comments (if provided):

Dear Dr. Meier,

Thank you for your manuscript submission.

Manuscript Number PDIG-D-21-00015 entitled "A proteomic survival predictor for COVID-19 patients in intensive care" has been reviewed. The comments of the reviewer(s) are included at the end of this letter.

There is interest in your manuscript, but as you can see from the reviews (below), substantial revisions are recommended and the paper must undergo further peer review. Therefore, I invite you to revise your manuscript, taking into account the suggestions made by the reviewers.

Reviewers' comments:

Reviewer's Responses to Questions

**Comments to the Author**

1. Does this manuscript meet PLOS Digital Health’s publication criteria? Is the manuscript technically sound, and do the data support the conclusions? The manuscript must describe methodologically and ethically rigorous research with conclusions that are appropriately drawn based on the data presented.

Reviewer #1: Yes

Reviewer #2: Partly

Reviewer #3: Yes

2. Has the statistical analysis been performed appropriately and rigorously?

Reviewer #1: Yes

Reviewer #2: I don't know

Reviewer #3: Yes

3. Have the authors made all data underlying the findings in their manuscript fully available (please refer to the Data Availability Statement at the start of the manuscript PDF file)?

Reviewer #1: No

Reviewer #2: Yes

Reviewer #3: Yes

4. Is the manuscript presented in an intelligible fashion and written in standard English?

Reviewer #1: Yes

Reviewer #2: Yes

Reviewer #3: Yes

5. Review Comments to the Author

Reviewer #1: Demichev et al. report the analysis of plasma proteomes from two cohorts of patients with severe COVID-19. Based on proteome profiles at early timepoints of maximum care the authors derive a predictor of clinical outcome for ICU patients, indicating that proteomics could outperform conventional risk assessment in terms of accuracy. This is a follow-up on a study just published in Cell Systems (ref 13) by the authors using the same data set, focusing on the specific point of predicting an individual’s outcome. As such it is of interest and I can recommend publication upon addressing the comments below.

1) Please indicate the size of the patient cohorts, number of samples and proteomics depth in the abstract as these are important for the interpretation of the statistical analysis.

2) The introduction states that molecular differences are less pronounced within the same than between different severity groups. Is there any data that could support this?

3) Please clarify the relation of the present manuscript to the study just published in Cell Systems. Figure 1 apparently recapitulates results from the Cell Systems study while the authors state that the longitudinal profile is only of limited value for the present study. Therefore, it would appear more appropriate to move panels b and c to Supplementary Material. Also, in the discussion: ‘Here, we show that an increase in specific inflammatory …’ Is this really a result of the present study?

4) It would be helpful to have a summary of the sample cohort akin to Suppl. Table 1 in Figure 1.

5) Please indicate how many proteins were included in the statistical model. I think it is important to discuss the potential of overfitting given the presumably large number of proteins as compared with the relatively small cohort size.

6) There is an increasingly large body of serum and plasma proteomics in COVID-19. Could the authors discuss their findings, e.g. the panel of inflammatory and acute phase proteins, more extensively in relation this this work? This would eventually help readers to assess the reproducibility of proteomic profiles.

7) It is customary to provide the underlying mass spectrometry raw data of proteomics studies. I recommend doing the same here or otherwise stating why this cannot be done and any restrictions that apply. Likewise, I would appreciate if the computer code could be shared via GitHub or similar.

Reviewer #2: The authors analyzed two small patient cohorts with repeated measurements over time for the prognostic purpose of predicting mortality in severely ill COVID-19 patients. Interestingly, they cite Wynants et al. (2020) but do not seem to follow the related advice in this living review. In particular, bold claims about clinical usefulness require strong evidence and should follow the advice given in the TRIPOD statement. However, this study suffers from several issues and I do not want to pick on the small number of events as this would be too easy and you would argue that proteomics is costly (in how far is this a re-analysis of already published data?). A bigger issue is the rather opaque machine learning part which will buy you additional flexibility resulting in overfitting. Also it is not clear to me if information from the future (beyond the cross-sectional time point at maximum treatment level) is included in the model - which would basically lead to a useless model for prediction purposes.

Reviewer #3: To the authors:

Dear Dr. Deminchev, dear Dr. Ralser and all appreciated colleagues,

Thank you for giving me the opportunity to review your interesting paper by Demichev and colleagues entiteled “A proteomic survival predictor for COVID-19 patients in intensive care” following submission to this appreciated journal.

At first, I wanna state that I do not have any conflict of interests and I did not find any concerns with plagiarism and self-plagiarism by the authors. Competing interests of some of the authors are clearly outlined.

Coronavirus disease 2019 (COVID-19) pandemia caused by SARS-CoV2 has affected millions of people worldwide with an unacceptable high mortality rate up to 60 % for critical cases. The course of the disease is highly variable, from asymptomatic course up to rapid deterioration in between hours with fulfilling criteria of severe sepsis. Some factors and parameters have been identified influencing or predicting severity and outcome, but early clinical parameters to delineate and allocate best treatment strategies are still not well established. However, these might be very helpful to distinguish the heterogeneous patient cohort in the emergency unit requiring medical attention and for estimation of associated mortality rate, especially at a time, when resources are limited due to a rise of severely ill patients.

For identification of proteomic parameters (obtained from serum) the authors aimed to identify (independent) biomarkers in two very well characterized longitudinal cohorts of patients with very similar medical treatment at the first time point fulfilling WHO-7 criteria (need for intensive care). Focusing on proteomic signatures is also gainful, since in COVID-19 patients changes on level of metabolites and small molecules are less pronounced due to less affection of cellular metabolic (dys-)function. At the end, a panel of 14 proteins is identified allowing prediction of survival at ICU-admission, which might quite helpful for clinical practice. The signature is similar, but not identical to these observed in other studies by Demichev (another paper of the first author), Geyer, Galbraight and Völlmy, supporting the concept, that proteomic analyses might improve prediction of clinical courses in COVID-19.

Congratulation to these impressive results. All in all, the paper is well written and consistent, nonetheless there are some questions and recommendations, which might improve the quality of the paper and should be handled prior to final decision of acceptance:

-. A short figure in supplementary data section about distribution (timely and individually) of the 321 proteins might increase clarity of the data.

- There is only a small body of information with respect of dealing with outliers.

- In the discussion section, a paragraph about structural and functional relationship (similarities and differences) would be helpful.

- Also in discussion section a paragraph on overall quality of identified biomarkers would be helpful – are these (un)specific danger signals during host response towards SARS-CoV2 or also other infectious agents as well as might they function as unspecific mortality markers?

6. PLOS authors have the option to publish the peer review history of their article (what does this mean?). If published, this will include your full peer review and any attached files.

**Do you want your identity to be public for this peer review?** For information about this choice, including consent withdrawal, please see our Privacy Policy.

Reviewer #1: No

Reviewer #2: No

Reviewer #3: No

---

## [Decision Letter · Decision Letter 1]

16 Nov 2021

PDIG-D-21-00015R1

A proteomic survival predictor for COVID-19 patients in intensive care

PLOS Digital Health

Dear Dr. Kurth,

Thank you for submitting your manuscript to PLOS Digital Health. After careful consideration, we feel that it has merit but does not fully meet PLOS Digital Health’s publication criteria as it currently stands. Therefore, we invite you to submit a revised version of the manuscript that addresses the points raised during the review process.

Specifically, please revise the text with regard to the translational potential toward clinical applications in light of the comments made by the Reviewer #2.

We look forward to receiving your revised manuscript.

Kind regards,

Martin G Frasch

Section Editor

PLOS Digital Health

Journal Requirements:

1. We notice that your supplementary figures are uploaded with the file type 'Figure' and are therefore included in the PDF. Please amend the file type to 'Supporting Information'. Please ensure that all Supporting Information files are included correctly and that each one has a legend listed in the manuscript after the references list.

Additional Editor Comments (if provided):

Dear Dr. Kurth,

Thank you for submitting your revised manuscript to PLOS Digital Health. Despite substantial improvement there is still concern on the interpretation of clinical applicability of the results.

Therefore, we invite you to submit a re-revised version of the manuscript that addresses the point mentioned above.

Reviewers' comments:

Reviewer's Responses to Questions

**Comments to the Author**

1. If the authors have adequately addressed your comments raised in a previous round of review and you feel that this manuscript is now acceptable for publication, you may indicate that here to bypass the “Comments to the Author” section, enter your conflict of interest statement in the “Confidential to Editor” section, and submit your "Accept" recommendation.

Reviewer #1: All comments have been addressed

Reviewer #2: (No Response)

Reviewer #3: All comments have been addressed

2. Does this manuscript meet PLOS Digital Health’s publication criteria? Is the manuscript technically sound, and do the data support the conclusions? The manuscript must describe methodologically and ethically rigorous research with conclusions that are appropriately drawn based on the data presented.

Reviewer #1: Yes

Reviewer #2: Yes

Reviewer #3: Yes

3. Has the statistical analysis been performed appropriately and rigorously?

Reviewer #1: Yes

Reviewer #2: No

Reviewer #3: Yes

4. Have the authors made all data underlying the findings in their manuscript fully available (please refer to the Data Availability Statement at the start of the manuscript PDF file)?

Reviewer #1: Yes

Reviewer #2: Yes

Reviewer #3: Yes

5. Is the manuscript presented in an intelligible fashion and written in standard English?

Reviewer #1: Yes

Reviewer #2: Yes

Reviewer #3: Yes

6. Review Comments to the Author

Reviewer #1: The authors have addressed the reviewers' points satisfactorily and revised their manuscript accordingly. I recommend publication.

Reviewer #2: Thank you for your comments. The authors do not seem to realize that the limiting sample size factor are the number of indiduals who died (i.e. 15 and 5!). Moreover, if only information from the cross-sectional time point is used for the prediction why is there all this additional processing involved (e.g. why parenclitic networks, did you or did you not impute values - again this is not clear to me but it is highly relevant when putting such an algorithm into practice). As a compromise I suggest to please tone down the text in the places where clinical applicability of the results is implied. The study as itself is interesting in particluar when it comes to biology.

Reviewer #3: The authors followed all of my recommendations and suggestions. I do not have further comments and recommend now acceptance of the paper in the revised version.

7. PLOS authors have the option to publish the peer review history of their article (what does this mean?). If published, this will include your full peer review and any attached files.

**Do you want your identity to be public for this peer review?** For information about this choice, including consent withdrawal, please see our Privacy Policy.

Reviewer #1: No

Reviewer #2: **Yes: **André Scherag

Reviewer #3: No

---

## [Editor Report · Decision Letter 2]

18 Nov 2021

A proteomic survival predictor for COVID-19 patients in intensive care

PDIG-D-21-00015R2

Dear Dr. Kurth,

We're pleased to inform you that your manuscript has been judged scientifically suitable for publication and will be formally accepted for publication once it meets all outstanding technical requirements.

Within one week, you'll receive an e-mail detailing the required amendments. When these have been addressed, you'll receive a formal acceptance letter and your manuscript will be scheduled for publication.

An invoice for payment will follow shortly after the formal acceptance. To ensure an efficient process, please log into Editorial Manager at https://www.editorialmanager.com/pdig/ click the 'Update My Information' link at the top of the page, and double check that your user information is up-to-date. If you have any billing related questions, please contact our Author Billing department directly at authorbilling@plos.org.

Kind regards,

Reinhard Bauer

Guest Editor

PLOS Digital Health